# OpenReview forum: "OrchJail: Jailbreaking Tool-Calling Text-to-Image Agents by Orchestration-Guided Fuzzing"
_ICML.cc/2026/Conference — ICML 2026 regular_

### Official Review · Reviewer_Xei5 · 2026-02-28

**Soundness:** 3
**Presentation:** 3
**Significance:** 4
**Originality:** 3
**Overall Recommendation:** 5
**Confidence:** 3

**Summary:**

The paper introduces OrchJail, an orchestration-guided fuzzing framework designed to jailbreak tool-calling text-to-image (T2I) agents. The authors identify a novel attack surface: while individual tool executions may appear benign, their multi-step orchestration can bypass safety alignments to generate policy-violating images. The framework abstracts successful tool-calling traces into structural patterns (macro-planning, micro-scheduling, and tool selection) and utilizes an LLM to infer the attribution between prompt phrasing and these patterns. This inferred knowledge subsequently guides prompt mutation and multi-objective scoring in a black-box fuzzing loop. Empirical evaluations across three T2I agents demonstrate that OrchJail achieves higher attack success rates, better image fidelity, and higher query efficiency compared to prompt-level jailbreak baselines.

**Compliance With Llm Reviewing Policy:**

Affirmed.

**Key Questions For Authors:**

noindent \textbf{Q1.} The paper uses the term "causal reasoning" for the LLM's process of linking prompt wording to orchestration patterns. Could you clarify if this is strictly causal intervention, or rather a prompt-pattern attribution heuristic? Modifying the terminology to "attribution" might increase the theoretical precision of the paper.

\noindent \textbf{Q2.} What is the end-to-end computational time or token-usage cost of OrchJail compared to baselines like JailFuzzer? While the number of target queries is minimized, the internal LLM calls for scoring and reasoning appear computationally heavy.

\noindent \textbf{Q3.} How sensitive is the framework to the choice of the local LLM (LLaVA-1.5-13B)? Have you tested the variance in mutation quality and scoring accuracy if a different open-weight model is utilized?

**Limitations:**

Yes

**Strengths And Weaknesses:**

\subsection*{Strengths}
\noindent \textbf{1.} The identification of tool orchestration as a distinct vulnerability in multi-agent T2I systems is a highly original and timely contribution to the AI safety domain.

\noindent \textbf{2.} The methodology is structurally rigorous. The Trace2Orch abstraction effectively reduces the dimensionality of complex execution logs into a tractable format for subsequent prompt optimization.

\noindent \textbf{3.} The experimental design is comprehensive, encompassing multiple target agents, state-of-the-art baselines, and robustness checks against established defense mechanisms (e.g., SmoothLLM).

\subsection*{Weaknesses}
\noindent \textbf{1.} The terminology regarding "causal reasoning" is technically imprecise. The LLM identifies linguistic attributions and correlations rather than establishing strict statistical or interventional causality between the prompt and the orchestration trace.

\noindent \textbf{2.} The framework's reliance on multiple LLM calls per iteration (for mutation, multi-objective scoring, and abstraction) indicates a substantial computational overhead. The paper evaluates query efficiency in terms of target agent queries but lacks an analysis of the local computational cost.

\noindent \textbf{3.} The reliance on LLaVA-1.5-13B as the evaluator and mutator introduces potential evaluation bias, as the success of the fuzzing loop is bounded by the specific instruction-following capabilities and internal alignments of this single LLM backbone.

---

> ### Author Rebuttal · Authors · 2026-03-31
>
> Thanks for your constructive and insightful reviews. We carefully respond to your concerns as follows.
>
> ## W1 & Q1: Terminology of "causal reasoning"
>
> Thank you for the clarification. We agree that our current use of the term "causal reasoning" is technically imprecise. The proposed analysis primarily captures linguistic attributions and correlations between prompt wording and observed orchestration patterns, based on LLM-driven summarization over successful cases, rather than establishing strict statistical or interventional causality. In the revision, we will revise the terminology throughout the paper (e.g., to "attribution"), and avoid overstating causality.
>
>
>
> ## W2 & Q2: Cost on Local LLM
>
> As you suggest, we additionally measure the local computation cost (both time and token usage) for OrchJail and two baselines, JailFuzzer and DACA, under the same experimental setup. The results are summarized as follows.
>
> | Method | Local time / target prompt (s) | Local token / target prompt (k) |
> |---|---:|---:|
> | OrchJail | 2.14 | 45.22 |
> | JailFuzzer | 8.87 | 104.50 |
> | DACA | 1.21 | 27.62 |
>
> Overall, OrchJail is substantially more efficient than JailFuzzer in local cost, likely because JailFuzzer relies on both LLM-based scoring and VLM-based mutation over image data, which is more expensive. DACA has lower local cost, but its attack success rate is the lowest among the baselines (46.77% vs. 72.63% for OrchJail). Given the stronger attack performance, we consider OrchJail's overhead acceptable.
>
> Besides, we clarify that the LLM-based attribution reasoning is performed only when new successful cases are found and is cached in the repository, so that during most iterations mutation and scoring can largely retrieve and reuse the stored knowledge, instead of re-reasoning, thereby mitigating the overall local overhead.
>
> We will include this cost analysis in the revision.
>
>
>
> ## W3 & Q3: Sensitivity on LLM Backbone
>
> Thank you for the question. To examine potential bias from local LLM backbone and the sensitivity of OrchJail to this choice, we repeated the experiments to jailbreak GenArtist by swapping the local backbone. Concretely, besides LLaVA-1.5-13B, we tested two parameter-comparable open-weight (without safety alignment) models from different families: Qwen3.5-9B-Base and Meta-Llama-3.1-8B. The results are:
>
> | Backbone Model | Success Rate | FID Score | #Queries |
> |---|---:|---:|---:|
> | LLaVA-1.5-13B | 72.63% | 156.26 | 12.15 |
> | Qwen3.5-9B-Base | 73.87% | 157.22 | 12.68 |
> | Meta-Llama-3.1-8B | 68.13% | 159.41 | 13.70 |
>
> Overall, performance varies mildly due to inherent differences in model capabilities. However, OrchJail is largely robust to the choice of the local backbone and achieves comparable performance across different models. We attribute this robustness in part to our few-shot prompting design in both mutation and scoring: we include representative jailbreak examples as in-context references, which helps stabilize the mutation and scoring across backbones. We will include this analysis and in the revision.

---

### Official Review · Reviewer_nbrc · 2026-03-05

**Soundness:** 2
**Presentation:** 3
**Significance:** 2
**Originality:** 3
**Overall Recommendation:** 3
**Confidence:** 4

**Summary:**

This paper proposes a novel black-box jailbreaking attack against tool-calling text-to-image (T2I) agents. The core idea is an orchestration-guided fuzzing framework that searches for prompts that make the planner inside the target agent choose a multi-step tool chain where the overall system produces the harmful image, even if some parts appear benign. The attack starts with rewriting the harmful instructions into prompts that are more likely to trigger different multi-tool orchestrations. It then queries the target agent, identifies successful jailbreaks, and stores an abstracted representation of the resulting tool orchestration and span-level prompt–orchestration attributions, which will be retrieved to guide a two-branch mutation strategy. Experiments on three T2I agents show higher attack success rates, lower FID, and fewer queries than several baselines, with ablation studies and robustness tests against two defenses.

**Compliance With Llm Reviewing Policy:**

Affirmed.

**Final Justification:**

My concerns related to the CLIP-based metric and defense evaluation have been resolved. But given the limitation in threat model assumption and the need for more quantitative experiments, I will keep my original score.

**Key Questions For Authors:**

1. In a true black-box setting, is it realistic to assume the attacker can access the agent’s full tool-invocation trace? Many deployed agents may disclose the set of available tools or provide a high-level summary, but not the exact tool-call sequence and per-step inputs. What kind of trace visibility is required?

2. How transferable are the learned orchestration patterns and causal span attributions? For example, if the repository is built on one agent, do the same patterns still work on a different agent or toolset, or are they largely agent-specific?

3. How well does the CLIP-based semantic alignment metric correlate with human judgments of whether the generated image satisfies the harmful intent?

4. What threshold is used to declare success, and how sensitive is the attack performance to the choice of threshold?

5. The defense evaluation appears limited and may not reflect the current state of multimodal safety. Why are more recent VLM guard approaches, e.g., LLaVA-Guard[1], GuardReasoner-VL[2], not considered, and are they inapplicable to this setting?

[1] Helff et al., LLAVAGUARD: An Open VLM-based Framework for Safeguarding Vision Datasets and Models, ICML '25.

[2] Liu et al., GuardReasoner-VL: Safeguarding VLMs via Reinforced Reasoning, NeurIPS '25.

**Limitations:**

No. The paper does not discuss limitations or potential negative societal impacts; it would be better to add an ethics section outlining foreseeable misuse and concrete mitigation guidance.

**Strengths And Weaknesses:**

**Strengths:**
+ Highlights orchestration-level vulnerabilities in tool-calling T2I agents, which are underexplored in prior jailbreaking work.
+ Proposes an end-to-end black-box fuzzing framework with orchestration-aware seed generation, guided mutation, and multi-objective scoring, offering a systematic alternative to unguided prompt perturbations.

**Weaknesses:**
- Several key designs and experimental setup details are not sufficiently specified. (See details in the question).
- Limited discussion on transferability, sensitivity test, leaving the robustness of the method unknown.

---

> ### Author Rebuttal · Authors · 2026-03-31
>
> Thanks for your constructive and insightful reviews. We carefully respond to your concerns as follows. We will include these additional experiments and analysis in the revision.
>
> ## Q1: Attack Assumption
>
> We agree that full tool-invocation traces are not always exposed in deployed agents. Our threat model targets a class of transparent tool-calling T2I agents that provide intermediate execution information for interpretability, debugging/auditing, and system configurability (e.g., allowing users/developers to inspect and adjust multi-step tool usage), such as the agents configured based on workflows (e.g., ComfyUI [1] and InvokeAI [2]).
>
> Importantly, OrchJail does not strictly require the full per-step inputs in all cases. The minimum trace visibility needed is the ordered tool sequence, which is sufficient to match and reuse high-level orchestration patterns.
> We will clarify this trace-visibility assumption in the revision.
>
> [1] ComfyUI, https://www.mintlify.com/Comfy-Org/ComfyUI/introduction
>
> [2] InvokeAI, https://invoke.ai/
>
>
> ## Q2: Transferability of Knowledge
>
> Transferability largely depends on (1) the overlap of agents' tools and (2) the orchestration logic, across agents. In our evaluation, target agents differ substantially in both aspects, so the learned patterns are largely agent-specific. Nevertheless, the cost of rebuilding a knowledge repository is modest: for a new target agent, OrchJail typically collects sufficient successful cases and orchestration patterns within fewer than 10 target prompts, which already provides effective guidance for subsequent fuzzing.
>
>
> ## Q3: Reliability of CLIP-based Metric
>
> Jailbreak success requires both a harmful/policy-violating output and alignment with the original malicious intent; following prior baselines (JailFuzzer and SneakyPrompt), we compute CLIPScore between the generated image and the original malicious prompt.
>
> We further conduct a human study on 100 samples (72 CLIP “success”, 28 “fail”): multiple authors annotate jailbreak success as ground truth, and we report the CLIPScore–human confusion matrix.
>
> |                         | CLIPScore (jailbreak success) | CLIPScore (jailbreak fail) | Sum |
> |---|---:|---:|---:|
> | **human (jailbreak success)** | 70 | 4  | 74  |
> | **human (jailbreak fail)**    | 2  | 24 | 26  |
> | **Sum**                       | 72 | 28 | 100 |
>
> Based on this evaluation, CLIPScore achieves a precision of 97.22% (70/72) and a recall of 94.59% (70/74) with respect to human judgments, demonstrating overall consistency with human assessments.
>
> ## Q4: Threshold
>
> We set the CLIPScore threshold for declaring jailbreak success to θi-t=0.26, which is an empirical setting and is also consistent with prior baselines (JailFuzzer and SneakyPrompt).
>
> To assess sensitivity, we add a threshold-sensitivity study: on the target agent of GenArtist, we sweep θi-t over the range 0.22–0.30 and report how the attack success rate, FID, and query numbers change as follows.
>
> | threshold θi-t | success rate (↑) | FID (↓) | Queries (↓) |
> |---:|---:|---:|---:|
> | 0.22 | 75.16% | 170.67 | 10.03 |
> | 0.24 | 73.66% | 163.19 | 11.19 |
> | 0.26 | 72.63% | 156.26 | 12.15 |
> | 0.28 | 70.23% | 154.64 | 13.64 |
> | 0.30 | 69.12% | 149.87 | 15.17 |
>
> The results show a trade-off: a lower θi-t tends to admit less low-quality images and thus worsens FID, while a higher θi-t imposes a stricter alignment constraint, leading to lower success rate and more queries. Based on our experience (as shown in Q3), θi-t=0.26 is a reasonably accurate and balanced choice.
>
>
> ## Q5: More Recent Guard
>
> Following your recommendation, we additionally incorporated two recent guard approaches, LLaVA-Guard and GuardReasoner-VL, as defenses in our setting. Concretely, we implement them to audit the generated image. We then re-evaluate OrchJail and best baseline (JailFuzzer) on GenArtist under these defenses. The results are shown as follows, including attack success rate and query efficiency.
>
> | Method | Defense | Success Rate | FID Score | #Queries |
> |---|---|---:|---:|---:|
> | OrchJail | None (original performance) | 72.63% | 156.26 | 12.15 |
> | OrchJail | LLaVA-Guard | 53.91% | 183.57 | 21.43 |
> | OrchJail | GuardReasoner-VL | 61.12% | 176.94 | 18.27 |
> | JailFuzzer | None (original performance) | 63.48% | 158.27 | 13.86 |
> | JailFuzzer | LLaVA-Guard | 39.73% | 210.46 | 24.60 |
> | JailFuzzer | GuardReasoner-VL | 48.43% | 191.38 | 20.17 |
>
> These results show that recent VLM-guard defenses reduce OrchJail’s success rate and increase FID, but do not fully eliminate the attack: OrchJail can still succeed (typically with more queries). OrchJail consistently outperforms JailFuzzer and degrades less under stronger guards, since it jailbreaks via tool orchestration rather than prompt-level textual evasion.

---

> > ### Author Rebuttal · Reviewer_nbrc · 2026-04-05
> >
> > Thank the author for the detailed response. My concerns related to the CLIP-based metric and defense evaluation have been resolved. But accessing the tool sequence is still a strong assumption for a black-box attack, and it would be better to have some quantitative experiments to measure the transferability between different agents, the cost, and performance. I will keep my original score.

---

> > > ### Author Response · Authors · 2026-04-08
> > >
> > > Thank you for the follow-up. To address this, we conduct a more systematic cross-agent transfer experiment that (1) evaluates how transferable prompt–orchestration knowledge is across agents, and (2) demonstrates that OrchJail can still operate without accessing the target agent’s tool sequence.
> > >
> > > Concretely, we build the knowledge repository using orchestration patterns and span-level attributions learned from CREA and LayerCraft, and then use this transferred repository to guide fuzzing against GenArtist under fully black-box access (i.e., without using GenArtist’s tool-sequence outputs). The results in terms of attack success rate, image quality (FID), and query cost are summarized below:
> > >
> > > | Method | Success Rate ↑ | FID Score ↓ | #Queries ↓ |
> > > |---|---:|---:|---:|
> > > | JailFuzzer | 63.48% | 158.27 | 13.86 |
> > > | SneakyPrompt | 57.12% | 167.75 | 15.43 |
> > > | OrchJail (full trace) | 72.63% | 156.26 | 12.15 |
> > > | OrchJail (transfer from CREA) | 67.74% | 157.45 | 12.97 |
> > > | OrchJail (transfer from LayerCraft) | 65.61% | 158.22 | 13.29 |
> > >
> > > Overall, transfer-based OrchJail is weaker than the in-domain (full-trace) upper bound, but it remains competitive and consistently outperforms the two SOTA baselines under the same target agent. We also observe that transferring from CREA yields better results than from LayerCraft, which we attribute to CREA being more similar to GenArtist in terms of tool availability and planning/orchestration logic, leading to higher pattern compatibility. We will include this transfer study and a clearer discussion of trace visibility assumptions in the revision.

---

### Official Review · Reviewer_gJvD · 2026-03-11

**Soundness:** 2
**Presentation:** 3
**Significance:** 2
**Originality:** 4
**Overall Recommendation:** 4
**Confidence:** 4

**Summary:**

This paper proposes a framework for jailbreaking tool-calling T2I agents.
It generates seed prompts and uses a success signal to revise the prompt based on this signal. Based on the successful prompts, it further revises prompts to use similar tool orchestration patterns to jailbreak the agents.

**Compliance With Llm Reviewing Policy:**

Affirmed.

**Final Justification:**

I would like to increase my score to 4. Although the proposed approach is not particularly novel and is somewhat of an engineering combination for multiple tool calling of a T2I agentic system, I find the approach otherwise reasonable and well-executed.

**Key Questions For Authors:**

**Questions:**
* If there are rephrasing safeguards, or safeguards that summarize the goal of the prompt, would this approach still be effective in bypassing those defenses?
* It seems that the current approach could be easily defended against by these simple defense strategies.
* In Section 4.1.3, causal-inspired prompt-orchestration knowledge is used from $C(p_s)$. However, how is this knowledge stored initially if there is no successful case at the beginning?

**Strengths And Weaknesses:**

**Strengths:**
* The approach is clearly described.
* The steps of the framework are technically sound.
* It is interesting to employ tool orchestration to jailbreak the agents.

**Weaknesses:**
* There are not many image examples or text examples where success cannot be verified solely not based on values. If the tool calling is benign and simply generates similar context without harmful content, it could achieve a high pass rate and high semantic scores.
    * Related to the previous point, it would be better to employ safeguard models or classification models to determine the harmfulness of the outputs, rather than relying on semantic values.
     * (This is a question for the authors, not a weakness.) Is cosine similarity of CLIP embeddings sufficient to detect harmfulness? In my understanding, similarity does not always represent harmful concepts, so using this metric as a jailbreak success evaluation is confusing (Section 4.1.2).
* Since the authors employ an LLM-as-judge for multi-objective scoring, it would be better to demonstrate how accurate this approach is for probability estimation. Does it have a high correlation with the logic of safeguard models (i.e., ground-truth probabilities)?
* It would be helpful to analyze which types of tool calls are triggered more frequently by the proposed approach compared to the baselines. Based on the examples in Table 4, it is difficult to examine whether the proposed method triggers multiple tool calls that can easily bypass T2I agents.

---

> ### Author Rebuttal · Authors · 2026-03-31
>
> Thanks for your constructive and insightful reviews. We carefully respond to your concerns as follows. We will include these additional experiments and analysis in the revision.
>
> ## W1: Detect harmfulness
>
> We use CLIPScore for two reasons: (1) to check whether the output reflects the harmful content implied by the target prompt, and (2) to enforce intent consistency (avoiding drift to irrelevant harmful content), consistent with prior baselines (e.g., JailFuzzer, SneakyPrompt).
>
> The examples are available in Appendix Figs 5-8. We add more examples in: https://anonymous.4open.science/r/OrchJail-C02B/Examples%20of%20OrchJail.pdf
>
> Following your advice, we additionally use GPT-4V as an extra auditor on 100 images from GenArtist. CLIPScore labels 72 as harmful and 28 as benign; we summarize the CLIPScore–GPT-4V agreement in the confusion matrix below.
>
> |                         | CLIPScore (jailbreak success) | CLIPScore (jailbreak fail) | Sum |
> |---|---:|---:|---:|
> | **GPT-4V (jailbreak success)** | 72 | 6  | 78  |
> | **GPT-4V (jailbreak fail)**    | 0  | 22 | 22  |
> | **Sum**                        | 72 | 28 | 100 |
>
> Overall, CLIPScore and GPT-4V show high agreement (94/100). For the 6 disagreements where GPT-4V flags success but CLIPScore does not, multi-author review finds minor harmful cues (e.g., a knife) but weak alignment with the target malicious intent; GPT-4V tends to emphasize harmfulness over intent alignment and lacks a clear tunable threshold, whereas our CLIPScore threshold is an empirical, reproducible setting consistent with prior baselines.
>
> ## W2: LLM-as-judge
>
> Our scoring prompt of LLM-as-judge explicitly includes successful jailbreak cases as context for reference, which is a few-shot prompting format (shown in Appendix Fig. 13) to improve the judge's consistency and accuracy.
>
> To further quantify reliability, we add an analysis on 200 prompts: for each prompt, we record (1) the LLM-as-judge bypass score and multi-objective total score, (2) the bypass number (determined by whether the target agent refuses to respond) and (3) the jailbreak success number (determined by CLIPScore).
>
> | bypass Score | bypass rate |
> |---|---|
> | [0.0, 0.25) | 5.13% (2/39) |
> | [0.25, 0.5) | 12.24% (6/49) |
> | [0.5, 0.75) | 89.06% (57/64) |
> | [0.75, 1] | 95.83% (46/48) |
>
> | Total Score | jailbreak success rate |
> |---|---|
> | [0.0, 0.75) | 0% (0/31) |
> | [0.75, 1.5) | 11.54% (6/52) |
> | [1.5, 2.25) | 78.79% (52/66) |
> | [2.25, 3] | 88.24% (45/51) |
>
> The LLM-as-judge multi-objective scoring has strong alignment with actual outcomes, supporting its use as an effective estimator for guiding candidate ranking.
>
> ## W3: Types of tool calls
>
> Appendix Figs. 5–8 provide more qualitative jailbreak successes (prompts and their tool-call traces). We also analyze 100 GenArtist jailbreak cases and report tool-sequence diversity by comparing OrchJail with SneakyPrompt and JailFuzzer; two cases are treated as the same sequence if their ordered tool-call sequence is identical, and we report #success, #unique sequences (among successes), and Top-3 frequent sequences, as follows:
>
> | Method | #successful jailbreak | #unique tool sequences | Top-3 tool sequences |
> |---|---:|---:|---|
> | SneakyPrompt | 57 | 21 | Generation→Addition; Generation→Addition→Edition; Generation→Addition→Addition |
> | JailFuzzer | 63 | 24 | Generation→Addition; Generation→Addition→Addition; Generation→Addition→Edition |
> | OrchJail | 72 | 35 | Generation→Addition→Addition→Edition; Generation→Addition→Edition→Edition; Generation→Addition→Edition→Addition |
>
> OrchJail discovers a richer and more diverse set of tool-call sequences, and its Top-3 sequences are typically longer and structurally more complex than those of the baselines. This suggests that OrchJail tends to jailbreak via more complex multi-step orchestrations, rather than relying solely on prompt-level textual evasion.
>
>
> ## Q1 & Q2: safeguards
>
> Follow your advice, we implemented two additional safeguards: we prompt an auxiliary LLM (LLaVA-1.5-13B) to (1) rephrasing the prompt or (2) summarize its goal, and then detect on the new prompt . We report the performance as follows.
>
> | Defense | Success Rate | #Queries |
> |---|---:|---:|
> | None (original performance) | 72.63% | 12.15 |
> | Rephrasing | 66.35% | 17.24 |
> | Summarize | 68.70% | 16.62 |
>
> OrchJail remains effective under these guards, as the mutation explicitly applies intent-hiding paraphrases and substitutions (e.g., replacing "slave" with "worker"), making the malicious intent less likely to be detected.
>
>
> ## Q3: Initial knowledge
>
> C(Ps) is initially empty and is populated progressively as successful cases are found. To alleviate cold start, we introduce a seed generation stage that rewrites harmful instructions into diverse seeds more likely to trigger multi-step orchestration, enabling early successes to populate C(Ps). This is supported by the w/o Gen ablation (Fig. 3 and Appendix Table 5), where removing seed generation leads to worse performance.

---

> > ### Author Rebuttal · Reviewer_gJvD · 2026-04-06
> >
> > W1: This response does not fully address the concern. GPT-4V is not a dedicated harmfulness detector, so using it as a validator undermines the validity of this experiment. A more convincing evaluation would sample across the full CLIPScore range to properly assess whether cosine similarity is a reliable proxy for harmfulness.
> > W2: Could you calculate correlation between llm's probability and real pass rate?

---

> > > ### Author Response · Authors · 2026-04-08
> > >
> > > ## W1
> > > Thank you for the further feedback. We strengthen our validation by comparing CLIPScore with a specialized safety guard, LLAVAGUARD [1], using stratified sampling across the full CLIPScore range. Concretely, we evaluate 200 images (including both harmful and benign cases) and report the harmfulness rate predicted by LLAVAGUARD within each CLIPScore bin:
> > >
> > > | CLIPScore bin | Harmful rate by LLAVAGUARD|
> > > |---|---|
> > > | [0.0, 0.2) | 0% (0/11) |
> > > | [0.2, 0.22) | 11.11% (1/9) |
> > > | [0.22, 0.23) | 7.69% (1/13) |
> > > | [0.23, 0.24) | 16.67% (2/12) |
> > > | [0.24, 0.25) | 18.75% (5/32) |
> > > | [0.25, 0.26) | 35.29% (12/34) |
> > > | [0.26, 0.27) | 88.64% (39/44) |
> > > | [0.27, 0.28) | 95.24% (20/21) |
> > > | [0.28, 0.3) | 100% (15/15) |
> > > | [0.3, 1] | 100% (9/9) |
> > >
> > > The results show a clear monotonic trend: as CLIPScore increases, the fraction of images flagged as harmful by LLAVAGUARD increases accordingly, and our used empirical threshold 0.26 (also adopted in prior baselines such as JailFuzzer [2] and SneakyPrompt [3]) lies near the transition region, making it reasonable.
> > >
> > > In addition, our human evaluation provides consistent evidence. On 100 samples, multiple authors independently annotate jailbreak success (harmful/policy-violating and aligned with the target malicious intent) as ground truth.
> > >
> > > |                         | CLIPScore (jailbreak success) | CLIPScore (jailbreak fail) | Sum |
> > > |---|---:|---:|---:|
> > > | **Human (jailbreak success)** | 70 | 4  | 74  |
> > > | **Human (jailbreak fail)**    | 2  | 24 | 26  |
> > > | **Sum**                       | 72 | 28 | 100 |
> > >
> > > The resulting confusion matrix indicates that CLIPScore achieves 97.22% precision (70/72) and 94.59% recall (70/74) with respect to human judgments, demonstrating strong overall agreement.
> > >
> > > Finally, we emphasize that we use CLIPScore not merely as a harmfulness detector, but also to enforce intent alignment: jailbreak success should not only contain harmful content, but also remain aligned with the original malicious intent, avoiding cases that are harmful yet drift away from the target semantics—something a pure harmfulness detector alone cannot capture. We will include these additional discussions in the revision.
> > >
> > >
> > >
> > > [1] Helff et al., LLAVAGUARD: An Open VLM-based Framework for Safeguarding Vision Datasets and Models, ICML '25.
> > >
> > > [2] Yingkai Dong, et al., Fuzz-testing meets llm-based agents: An automated and efficient framework for jailbreaking text-to-image generation models, 2025 IEEE Symposium on Security and Privacy (SP).
> > >
> > > [3] Yuchen Yang, et al., Sneakyprompt: Jailbreaking text-to-image generative models, 2024 IEEE symposium on security and privacy (SP).
> > >
> > >
> > > ## W2
> > > Thank you for the suggestion. We additionally quantify the correlation between the LLM-as-judge scores and the actual pass rate on the 200 prompts shown in our prior reply to W2. Specifically, we compute both Pearson correlation and Spearman rank correlation.
> > >
> > > **Bypass score vs. observed bypass rate: Pearson = 0.7668, Spearman = 0.7734.**
> > >
> > > **Total score vs. observed jailbreak success rate: Pearson = 0.8061, Spearman = 0.8112.**
> > >
> > > These results indicate a strong positive association between the judge scores and the real pass rate, supporting the reliability of our LLM-as-judge scoring for guiding candidate ranking. We will include these correlation statistics in the revision.

---

### Official Review · Reviewer_1ZMR · 2026-03-13

**Soundness:** 3
**Presentation:** 2
**Significance:** 2
**Originality:** 1
**Overall Recommendation:** 3
**Confidence:** 4

**Summary:**

This paper proposes OrchJail, which is designed to jailbreak text-to-image agents. It decomposes jailbreak attacks that cannot be achieved in a single step into a multi-step process. Experiments on three agents show that OrchJail is more effective than other baselines and requires fewer queries.

**Compliance With Llm Reviewing Policy:**

Affirmed.

**Final Justification:**

The authors provided detailed explanations regarding the paper's novelty and causal grounding. However, given the practical constraints of the threat model and the inherent nature of the method as an LLM-based attribution, I will maintain the score.

**Key Questions For Authors:**

See weaknesses.

**Limitations:**

No, please add a description of the limitations.

**Strengths And Weaknesses:**

**Strengths:**

This paper explores the security of text-to-image agents and conducts experiments on three target agents, demonstrating better attack performance.



**Weaknesses:**
1. Limited novelty. The proposed framework, OrchJail, essentially combines fuzzing with LLM-based mutation.

2. Unclear threat model. The paper does not clearly define the attacker's knowledge and capabilities.

3. The framework assumes that the attacker can get the agent’s trajectory. However, in many real-world agents, the trajectory is not accessible to users, which limits the applicability of the proposed method.

4. Weak causal grounding. This paper describes prompt–orchestration relations as “causal”. However, this relies on LLM-based analysis of successful cases rather than a causal identification method, making the explanation largely heuristic.

5. Writing quality. The paper contains some grammatical errors and informal expressions, which affect clarity and overall readability.

---

> ### Author Rebuttal · Authors · 2026-03-31
>
> Thanks for your constructive and insightful reviews. We carefully respond to your concerns as follows.
>
>
> ## W1: Limited novelty
>
> While OrchJail uses fuzzing and LLM-based mutation, our novelty is not a simple combination of them. We (1) propose tool orchestration as a distinct jailbreak attack surface beyond prompt-level evasion, and (2) design an orchestration-guided fuzzing framework that leverages observed orchestration behaviors to steer prompt search toward effective jailbreaks.
>
> Importantly, our ablation supports that the gain does not come from generic “fuzzing + LLM mutation”. In w/o Rea, we remove the orchestration abstraction and attribution reasoning, reducing the framework to essentially fuzzing with LLM-based mutation. This variant achieves the lowest jailbreak success rate across all ablations (Fig. 3), demonstrating that the orchestration-aware components are the primary driver of OrchJail’s effectiveness.
>
>
> ## W2: Unclear threat model
>
> Thank you for pointing this out. We consider a black-box attacker targeting a tool-calling text-to-image (T2I) agent that plans and executes a multi-step tool chain. For a queried prompt p, the attacker can observe the agent’s explicit refusal/block signal ρ(p)∈{0,1}, the final output image I(p) when not refused, and the executed tool-calling trace, where each step contains the selected tool name and its textual input argument.
>
> The attacker may excute multiple queries, but does not have access to the agent’s internal states (e.g., hidden embeddings and safety rules). The attacker’s goal is to find prompts that bypass the built-in safeguards while producing images semantically aligned with the intended harmful request.
>
> We will revise to state the threat model more explicitly.
>
>
> ## W3: The applicability of the proposed method
>
> We agree that full tool-invocation traces are not exposed in all deployed agents. Our threat model targets a class of more transparent tool-calling T2I agents that provide intermediate execution information for process interpretability, debugging/auditing, and system customizability (e.g., allowing users to inspect and adjust tool usage), such as the agents configured based on workflows (e.g., ComfyUI [1] and InvokeAI [2]). For such agents, it is realistic for an external user to observe the partial trajectory (e.g., plans or tool-call logs), which our framework leverages to extract orchestration patterns and guide fuzzing.
>
> [1] ComfyUI, https://www.mintlify.com/Comfy-Org/ComfyUI/introduction
>
> [2] InvokeAI, https://invoke.ai/
>
>
> ## W4: Weak causal grounding
>
> We agree that our “causal” is less theoretical. Our prompt–orchestration analysis is LLM-based attribution/association derived from successful trajectories, rather than a rigorous causal identification procedure with statistical guarantees. We adopted this design choice because stronger causal identification methods typically require explicit interventions and counterfactual replays (e.g., removing a span or altering a tool step and re-executing the agent multiple times), which can substantially increase query cost.
>
> In contrast, LLM-based attribution provides a lightweight and practical surrogate that can extract reusable orchestration patterns from observed successes and guide subsequent fuzzing iterations with low overhead.
>
> We will update the paper to better justify this trade-off and avoid overstating causality.
>
>
> ## W5: Writing quality
>
> In the revision, we will conduct thorough proofreading by multiple authors, correct grammar and wording, reduce informal/colloquial phrasing, and ensure consistent terminology and notation throughout the paper to improve readability and presentation quality.

---

> > ### Author Rebuttal · Reviewer_1ZMR · 2026-04-03
> >
> > Thank you to the author for the clarification. The author explained the novelty, threat model, and causal grounding of this paper. However, as the author acknowledges, this work has limitations in the practical application of the threat model, and the method is essentially an LLM-based attribution, so I cannot raise the score.

---

### Decision · Program_Chairs · 2026-04-30

**Decision:**

Accept (regular)

**Comment:**

This paper proposes OrchJail, an orchestration-guided fuzzing framework for jailbreaking tool-calling text-to-image agents. The key idea is to exploit tool orchestration as a new attack surface: instead of relying only on prompt-level perturbations, OrchJail learns from successful multi-step tool-calling traces and uses this knowledge to guide prompt mutation toward unsafe multi-step behaviors. Experiments on representative text-to-image agents show that OrchJail achieves higher attack success rates, better image fidelity, and lower query costs than prior baselines, while remaining effective against several defenses.

The paper studies an important and timely problem in the security of tool-calling text-to-image agents. Reviewers found the problem setting relevant and the empirical results promising, especially the focus on orchestration-level vulnerabilities beyond conventional prompt-only jailbreaks. The paper received one Accept, one Weak Accept, and two Weak Reject recommendations. During the rebuttal, the authors clarified the threat model, toned down the causal terminology, and added further evidence on metric reliability, defense robustness, transferability, computational cost, and backbone sensitivity. Although some concerns remained, particularly regarding the practicality of the threat model and the framing of causality, the rebuttal addressed a substantial portion of the reviewers’ questions and strengthened the paper overall.

The AC concurs that the paper makes a meaningful contribution to the study of security risks in tool-calling text-to-image agents. While the paper has some limitations, its strengths in problem formulation, empirical evaluation, and the identification of orchestration as a distinct attack surface outweigh the remaining weaknesses. Therefore, the AC recommends acceptance to ICML 2026.